# UPLC-G2Si-HDMS Untargeted Metabolomics for Identification of Yunnan Baiyao’s Metabolic Target in Promoting Blood Circulation and Removing Blood Stasis

**DOI:** 10.3390/molecules27103208

**Published:** 2022-05-17

**Authors:** Qingyu Zhang, Aihua Zhang, Fangfang Wu, Xijun Wang

**Affiliations:** 1National Engineering Laboratory for the Development of Southwestern Endangered Medicinal Materials, Guangxi Botanical Garden of Medicinal Plant, Nanning 530000, China; zhangqingyu926@163.com (Q.Z.); wffttn@163.com (F.W.); 2National Chinmedomics Research Center, National TCM Key Laboratory of Serum Pharmacochemistry, Metabolomics Laboratory, Department of Pharmaceutical Analysis, Heilongjiang University of Chinese Medicine, Heping Road 24, Harbin 150040, China; aihuatcm@163.com; 3State Key Laboratory of Quality Research in Chinese Medicine, Macau University of Science and Technology, Avenida Wai Long, Taipa 999078, Macau

**Keywords:** Yunnan Baiyao, biomarker, metabolic pathway, metabolomics, UPLC-G2Si-HDMS

## Abstract

Yunnan Baiyao is a famous Chinese patent medicine in Yunnan Province. However, its mechanism for promoting blood circulation and removing blood stasis is not fully explained. Our study used metabonomics technology to reveal the regulatory effect of Yunnan Baiyao on small molecular metabolites in promoting blood circulation and removing blood stasis, and exploring the related urine biomarkers. The coagulation function, blood rheology, and pathological results demonstrated that after Yunnan Baiyao treatment, the pathological indexes in rats with epinephrine hydrochloride-induced blood stasis syndrome improved and returned to normal levels. This is the basis for the effectiveness of Yunnan Baiyao. UPLC-G2Si-HDMS was used in combination with multivariate statistical analysis to conduct metabonomic analysis of urine samples. Finally, using mass spectrometry technology, 28 urine biomarkers were identified, clarifying the relevant metabolic pathways that play a vital role in the Yunnan Baiyao treatment. These were used as the target for Yunnan Baiyao to promote blood circulation and remove blood stasis. This study showed that metabolomics strategies provide opportunities and conditions for a deep and systematic understanding of the mechanism of action of prescriptions.

## 1. Introduction

Yunnan Baiyao was founded in 1902 and is a well-known traumatic medicine in China. Clinical studies found that Yunnan Baiyao can improve blood microcirculation obstacles, promote capillary growth, reduce the risk of thrombosis, inhibit the release of inflammatory mediators, and increase capillary permeability during the development of inflammation to achieve better blood circulation. It could also remove blood stasis [1,2], stop bleeding [3,4,5], relieve pain, be anti-inflammatory, and reduce swelling [6,7,8]. There are a variety of dosage forms, such as aerosols, powders, tablets, and capsules. It is widely used in treating various bleeding and blood stasis diseases both in internal medicine and external medicine, especially in curing/preventing women’s, and children’s diseases, in curing/preventing five sense organs, and skin diseases [9,10], all of which achieved satisfactory therapeutic effects.

Modern research believes that blood stasis syndrome is related to blood microcirculation disorders, abnormal blood flow distribution in internal organs and limbs, blood coagulation, abnormal hemorheology, immune dysfunction, abnormal connective tissue metabolism, and pathological inflammatory processes [11,12]. With the continuous research on blood stasis treatment and clinical practice, many studies showed that blood stasis syndrome is closely related to blood stasis physique. Blood stasis syndrome can manifest various clinical diseases, including internal or external bleeding; in women and children; in diseases of the skin, the five sense organs, nerves, and even tumors. Blood stasis causes a wide range of diseases [13]. Therefore, research on blood stasis syndrome and promoting blood circulation to remove blood stasis has been the most active field in traditional Chinese medicine and integrated traditional Chinese and Western medicine. At present, the research on the pathogenesis of blood stasis syndrome has gradually deepened to a molecular level [14]. However, due to the complex characteristics of multi-component and multi-target prescriptions, the mechanism of traditional Chinese medicine prescriptions to promote blood circulation and remove blood stasis is lacking.

Metabolomics is the overall analysis of all metabolites in the organism and the relationship between metabolites and physiological and pathological changes. It is an integral part of systems biology. Metabolomics technology can quickly and accurately analyze metabolites, identify biomarkers and directly analyze metabolomics in traditional Chinese medicine (TCM) clinical syndrome diagnosis and study the mechanism and performance of traditional Chinese medicine [15,16]. It is a widely used method. In this study, a non-targeted metabolomics method based on high-resolution liquid chromatography-quadrupole time-of-flight mass spectrometry (UPLC-QTOF-MS) was established to study the endogenous metabolism of Yunnan Baiyao in rat urine. Potential metabolic markers related to blood stasis syndrome were screened, and the metabolic pathways and in vivo mechanism of action related to drug differences were analyzed. The methodological reference for the study of Yunnan Baiyao’s mechanism on promoting blood circulation and removing blood stasis and its scientific basis was provided using multivariate statistical analysis methods to analyze and compare the three groups of samples.

## 2. Materials and Method

### 2.1. Drugs and Chemical Reagents

Adrenaline hydrochloride injection (1 mg/mL, batch number: H12020526) was purchased from Tianjin Jinyao Pharmaceutical Co., Ltd. (Tianjin, China). Yunnan Baiyao manufacturer provided Yunnan Baiyao samples; Pentobarbital sodium powder was purchased from Shanghai Chemical Reagent Procurement and Supply Station (Shanghai, China). The reagents acetonitrile, methanol, formic acid, leucine-enkephalin, used in ultra-high-performance liquid chromatography-mass spectrometry, were of high-performance liquid chromatography grade, and other reagents used were of analytical grade.

### 2.2. Animals and Experimental Design

Sprague Dawley (SD) male rats (9 weeks old) were purchased from the Experimental Animal Center of Heilongjiang University of Traditional Chinese Medicine (Harbin, China), certificate number: SCXK (Liao) 2015-0001. SD rats were raised in an environment at 25 ± 1 °C, with 50 ± 5% humidity, and 12 h/12 h light/dark cycles approved by the International Laboratory Animal Management Evaluation and Certification Association (Chicago, IL, USA), and with free access to eating and drinking. All rats were divided randomly into two control groups (CON, Control), two model groups (MOD, Model), and one treatment group (YNBY), with 11 rats in each group. Except for the control group, the rats in each group were administered a subcutaneous injection of epinephrine hydrochloride (0.3 mg/kg) for seven consecutive days daily. The CON and MOD groups were sacrificed on the 8th day for model evaluation. The treatment group was given an equivalent dose of Yunnan Baiyao solution (180 mg/kg) by gavage on the 8th day. The other groups of rats were given distilled water daily by gavage for 15 consecutive days. On the 23rd day, the Control, Model, and YNBY groups were sacrificed for efficacy evaluation. The research was conducted following the principles stated in the Declaration of Helsinki and approved by the Ethics Committee of Heilongjiang University of Traditional Chinese Medicine (Harbin, China).

### 2.3. Hemorheology, Coagulation Function, and Histopathology Detection Methods

The Z30 automatic blood rheology tester (Nanjing Zhilun Technology Co., Ltd., Nanjing, China) and the CA-7000 automatic blood coagulation tester (SYSMEX, Japan) were self-checked following the manufacturer’s instructions before use. The measurement can be carried out only after the instrument’s parameters are normal and the quality control standard is within the normal range. The measured value was valid. The rat heart, liver, spleen, lung, and kidney tissues were collected for hematoxylin (HE) staining and observed under a light microscope. Image analysis was performed using Image-Pro Plus 5.0 software (Media Cybernetics, Rockville, MD, USA).

### 2.4. Urine Collection and Preparation

Before the experiment, all animals were acclimated to metabolic cages for three days. Urine samples of rats were collected in the metabolic cage at 8 am on the 0, 1, 3, 5, and 7 days. The samples were centrifuged at 4 °C at 13,000 rpm for 15 min. The supernatant was diluted in HPLC-grade ultrapure water (1:7 dilution), vortexed for 10 s, centrifuged (4 °C, 13,000 rpm, 15 min), passed through a 0.22 µm microporous membrane, and used for UPLC-QTOF-HDMS analysis.

### 2.5. Metabolomics Analysis

The chromatographic analysis was conducted on Waters Acquity^TM^ Ultra Performance LC System (Waters, Milford, CT, USA), equipped with a quaternary pump, vacuum degasser, autosampler, and diode array detector. The analysis consisted of ACQUITY UPLC^TM^ HSS T_3_ (Waters; Milford, CT, USA) 1.8 µm 2.1 × 100 mm column, 2 µL injection volume, and 0.4 mL/min flow rate. Mobile phase A was 0.1% formic acid-acetonitrile, and mobile phase B was 0.1% formic acid-water. Gradient elution program: 0–2.5 min, 1–11% A; 2.5–4.5 min, 11–21%; 4.5–7 min, 21–40%; 7–8.5 min, 40–99% A. The mass spectrometry system used was a quadrupole time-of-flight mass spectrometer (Synapt G2-Si HDMS, Waters Corporation, Wilmslow, MA, USA), equipped with an ESI ion source, in the full scan mode with 50–1200 *m*/*z* positive ionization mode and negative ionization mode. ESI^+^ parameters: 3.0 kV capillary voltage; 30 V cone voltage; the dissolvent gas temperature at 350 °C; 800 L/h dissolvent gas flow rate; ion source temperature at 110 °C. The accurate mass calibration used leucine-brain for the orphanin ([M + H]^+^ = 556.2771) solution, and the sampling speed of the calibration solution was 10 µL/min, with 0.2 s calibration frequency, and the workstation adopted MassLynx V4.1 workstation (Waters, Milford, USA). ESI^−^ parameters: 2.5 kV capillary voltage; cone voltage: 20 V; calibration fluid leucine-enkephalin ([M − H]^−^ = 554.2615); other parameters were the same as positive ions. The effluent from the chromatograph was directly injected into the mass spectrometer without being split for positive and negative ion scanning analysis.

### 2.6. Multivariate Statistical Analysis

Rat urine metabolomics data were processed and analyzed by Progenesis^®^ QI software processing system (Nonlinear Dynamics, Newcastle, UK). The between-subject design gives each sample grouping information, and the chromatographic peaks in the sample were analyzed. Preprocessing included peak extraction, peak denoising, peak matching, peak alignment, and normalization. The obtained data matrix was imported into Ezinfo3.0 software (Waters, Milford, CT, USA) for multivariate statistical analysis, including principal component analysis (PCA) and orthogonal partial least squares discriminant analysis (OPLS-DA). The VIP map was established based on OPLS-DA, and the different variables that have an important contribution to the classification (VIP > 1) were found from the dataset. Simultaneously, combined with the results of the *t*-test (*p* < 0.05) between the groups, the key metabolites were delineated.

### 2.7. Metabolite Identification

According to the retention time (Rt) and accurate mass (*m*/*z*) of key metabolites provided by UPLC-MS, element composition analysis (resolution > 10,000; precision < 5 ppm) was used to determine the possible chemical equation (formula). The Human Metabolome Database (HMDB, http://www.hmdb.ca accessed on 6 May 2019), Metlin (http://masspec.scripps.edu/ accessed on 20 July 2019), and other metabolites and mass spectrometry databases were searched using accurate mass, possible chemical equation, and MS/MS data as clues. Mass spectrometry information matching was performed to infer the possible attribution of metabolites preliminarily. Further to this, the matched compounds and corresponding secondary mass spectrometry information were input into MassLynx-nested Massfragment software (Waters, Milford, CT, USA), according to the chemical structure. The possibility of cleavage and the cleavage law of biological mass spectrometry verified the validity of the identification.

### 2.8. Construction and Analysis of Biological Network of Key Biomarkers

The key metabolites regulated by Yunnan Baiyao were imported into MetaboAnalyst5.0 (https://www.metaboanalyst.ca/ accessed on 25 September 2019) to create a metabolite–gene network and identify metabolite-related genes. To find disease genes, the key words “Hemorheological abnormality” and “Coagulopathy” were entered into the GeneCards (https://www.genecards.org/ accessed on 26 September 2019) database, and the Venn diagram was used to determine the intersection of metabolite-related genes and disease genes. The common genes obtained from the Venn diagram were imported into the STRING (https://string-db.org/ accessed on 26 September 2019) online database for analysis, and a protein–protein interaction (PPI) network was generated. The network was imported into the Cytoscape 3.8.0 software (https://cytoscape.org/ accessed on 28 September 2019). The Network Analyzer tool was used to analyze topology and select the genes with degrees of freedom greater than the average value as potential targets of Yunnan Baiyao for promoting blood circulation and removing blood stasis. The obtained core targets were subjected to combined pathway analysis using the Network Analyst section of the MetaboAnalyst5.0 (https://www.metaboanalyst.ca/ accessed on 21 October 2019) website to analyze the relationship between targets and pathways.

### 2.9. Statistical Analysis

The statistical data were processed using GraphPad Prism 8.3.0 software (GraphPad Software, San Diego, CA, USA), and the statistical results were displayed as a bar graph. The comparison between groups was performed using a *t*-test, and *p* < 0.05 was regarded as statistically significant.

## 3. Results

### 3.1. Hemorheology, Blood Coagulation Function Test, and Histopathology Test

Epinephrine hydrochloride was used to replicate the blood stasis syndrome model in rats. After seven consecutive days, compared with the control group, the rats in the model group showed signs of reduced mobility, being curled up and moving less, being drowsy, and a dull coat. The hemorheology results showed that the plasma viscosity of the model rats increased, the whole blood viscosity increased, and the shear rate was 3 (1/s), 30 (1/s), 50 (1/s), and 100 (1/s). The difference was highly significant (*p* < 0.01). When the shear rate was 180 (1/s), the difference was statistically significant (*p* < 0.05) (Figure 1a–f). The coagulation function test results showed that the model group had large Rat plasma thrombin time (*p* < 0.01), plasma prothrombin time (*p* < 0.05) was shortened, and fibrinogen content increased (Figure 1g–i). Pathological results showed that the heart, liver, spleen, lungs, and kidneys of model rats had different degrees of vasodilation, blood stasis, edema, and other pathological changes in blood stasis syndrome (Figure 2a–e), of which the changes in the heart and spleen were most significant. Thus, indicating that the model was copied successfully. Yunnan Baiyao was continuously treated for 15 days based on the successful replication of the blood stasis rat model. Compared with the model group, the appearance and behavior of the rats in the administration group improved; specifically, the mental state and activity level were improved and tended to normal. The viscosity of rat plasma and whole blood decreased. When the shear rate was 100 (1/s), the difference was highly significant (*p* < 0.01), and when the shear rate was 180 (1/s), the difference was statistically significant (*p* < 0.05) (Figure 1j–o). Moreover, the rat plasma thrombin time and plasma prothrombin time was prolonged (Figure 1p–r). Yunnan Baiyao effectively improved the heart, liver, and spleen of rats with blood stasis syndrome, lung and kidney stasis, congestion, edema, and inflammatory cell infiltration, and other pathological condition, so that the organs of the rat returned to the level close to that of the normal rat (Figure 2f–j). The heart, liver, and kidney stasis treatment effect were good, indicating that Yunnan Baiyao prescription had a good treatment effect on blood stasis syndrome.

### 3.2. Metabolite Profile Analysis and Biomarker Identification

The urine samples of the control group, model group, and Yunnan Baiyao group were collected in positive and negative ion mode to obtain the corresponding BPI chromatogram using the established metabolomics method (Figure 3). The metabolic data were imported into Progenesis^®^ QI (Waters, Milford, USA). The software system was preprocessed and imported into the Ezinfo 3.0 software (Waters, Milford, USA) to perform unsupervised PCA analysis on the rat urine metabolism data on the 0, 1, 3, 5, and 7 days during the replication of the blood stasis model. The study found that the model group’s metabolism trajectory gradually deviated from the control group (Figure 4a,f) over time, and then PCA and OPLS-DA analysis were performed on the metabolic data on the 7th day of the modeling. The changes in the metabolic trajectory after the modeling were observed to obtain the corresponding score plot (Figure 4b,c,g,h). It can be seen from the figure (Figure 4b,c,g,h) that the metabolic trajectories of the blood stasis syndrome model group and the control group are completely separated, indicating that the metabolic network in rats changed significantly after administration of epinephrine hydrochloride for 7 days. Subsequently, S-plot and VIP-plot diagrams (Figure 4d,e,i,j) directly reflect the critical influencing factors on the generated metabolic profile. In the S-plot and VIP-plot diagrams, the farther the ion from the center contributes to the dispersion between groups. The scatter points with VIP > 1 were selected, and the between-group *t*-test was used to filter out those with a *p*-value < 0.05. Ions were used as potential biomarkers of blood stasis syndrome. Then, the 28 potential metabolic biomarkers of urine in the blood stasis rat model were finally characterized by matching the cleavage information of the secondary fragment mass spectrometry of the candidate biomarkers (Table 1). The Yunnan Baiyao group has a 22/28 callback trend (Figure 5). The results of the secondary ion fragmentation information (Appendix A) matching of these 28 compounds are shown in the Appendix A, using the “Statistical Analysis” module in the MetaboAnalyst 5.0 online analysis software to generate a visual heat map (Figure 6) that visually evaluates the similarity between samples and analyzes the dynamic changes in 28 urine biomarkers. The hue of the block reflects the relative signal strength and cluster recognition. The variables in the control and model groups were significantly different, indicating that the metabolites in the blood stasis syndrome rats had significant changes. The metabolic profile of the Yunnan Baiyao group was close to the control group, indicating the effectiveness of Yunnan Baiyao in treating blood deficiency syndrome.

### 3.3. Analysis of Related Metabolic Pathways of Serum Biomarkers

The 28 selected urine potential biomarkers of blood stasis syndrome were imported into MetaboAnalyst 5.0 online analysis software for metabolic pathway analysis. Eight related metabolic pathways were obtained, including tyrosine metabolism, pyrimidine metabolism, taurine, and sub-Taurine metabolism, phenylalanine metabolism, histidine metabolism, lysine degradation, arachidonic acid metabolism, and tryptophan metabolism (Figure 7). The results showed that small-molecule endogenous metabolites had a strong interference effect in the entire metabolic process, and this performance is closely related to blood stasis syndrome (Figure 8). These metabolic disorders caused by model replication can be used to explain the pathogenesis of blood stasis syndrome.

### 3.4. Construction and Analysis of Biological Network of Key Biomarkers

The key metabolites regulated by Yunnan Baiyao were chosen for biological network analysis, and their potential targets for promoting blood circulation and removing blood stasis were investigated using MetaboAnalyst5.0. The keywords “Hemorheological abnormality” and “Coagulopathy” were entered into the GeneCards website to search for disease gene targets, and a total of 102 species were obtained. A Venn diagram revealed 40 common targets at the intersection of metabolite-related genes and disease-related genes (Figure 9). The obtained common targets were imported into the STRING online database for analysis; the search was restricted to species “homo sapiens,” and independent nodes were kept hidden with the highest confidence (highest confidence) = 0.900 as the standard, and a PPI network diagram was created. The protein interaction relationship was imported into Cytoscape3.8.0 software, topological analysis was performed using the Network Analyzer tool, the genes with a degree of freedom greater than 3.071 were identified as potential targets, and F2, ALOX15, EDN1, MAPK14, IL4, HGF, IL6, PTGS2, MAPK3, and PLA2G4A were identified to play significant roles in the network graph (Figure 10). Through a review of the literature, Yunnan Baiyao’s core targets for promoting blood circulation and removing blood stasis can be identified. Prothrombin encoded by F2 is proteolyzed to form activated serine protease thrombin. During thrombus formation and hemostasis, activated thrombin converts fibrinogen to fibrin, causing platelet aggregation by activating other coagulation factors. Thrombosis and prothrombinemia can be caused by mutations in this gene [17,18,19]. ALOX15 encodes a non-heme iron-containing dioxygenase that catalyzes the production of a number of bioactive lipid mediators [20,21,22,23]. The 12-hydroperoxyeicosatetraenoate/12-HPETE and 15-hydrogen Peroxyeicosatetraenoate/15-HPETE resulted from the insertion of a peroxy group at C12 or C15 of arachidonic acid ester ((5Z,8Z,11Z,14Z)-eicosatetraenoate) [20,21]. It has pro-inflammatory properties and has been thought to act on 12-HPETE to produce hepoxilins. It may participate in the sequential oxidation of (4Z, 7Z, 10Z, 13Z, 16Z, 19Z) docosahexaenoate to produce pro-decomposition mediators (SPM), which have anti-platelet aggregation and immune-regulatory properties [23]. It has been shown to be involved in cellular responses to IL13/interleukin-13 [24]. EDN1 encodes for a preproprotein that is hydrolyzed to produce endothelin. Endothelin is made up of highly effective vasoconstrictor peptides. It is a potent vasoconstrictor, and endothelin antagonists can help treat certain cardiac, vascular, and renal diseases [25]. The obtained core targets were subjected to a combined pathway analysis of “gene-metabolite-metabolic pathway” using the Network Analyst section of the MetaboAnalyst5.0 (https://www.metaboanalyst.ca/ accessed on 21 October 2019) website. Our results indicate that “ALOX15, EDN1- Prostaglandin E2-arachidonic acid metabolism” may be an important pathway for Yunnan Baiyao to promote blood circulation and remove blood stasis. 

## 4. Discussion

Yunnan Baiyao is one of the famous traditional Chinese medicines in traumatology, with a history of clinical application for more than 100 years. However, there are few reports on its specific efficacy and mechanism, especially the mechanism of promoting blood circulation and removing blood stasis from the perspective of metabonomics. Metabolomics is an emerging strategy based on the study of the differences in the body’s metabolite profiles to discover the target points of Chinese herbal compound prescriptions and explain their overall mechanism [26,27]. Given the consistency between the characteristics of integrated metabolomics analysis and the thinking mode of the TCM holistic view, it is feasible to use metabolomics to reveal the pathological nature of TCM syndromes and the effect mechanism of TCM compound prescriptions [28,29]. Predicting biomarkers or molecular targets and metabolic pathways is conducive to diagnosing, finding a prognosis, and treating syndromes or diseases [30,31,32,33,34,35]. The emergence of some metabolite databases and metabolic pathway analysis software [36,37,38] provides an important basis for mining and identifying biomarkers.

This experiment used a long-term subcutaneous injection of low-dose epinephrine hydrochloride to simulate the chronic Qi stagnation and blood stasis syndrome, reflecting TCM syndromes’ dynamic temporal and spatial characteristics. Using classic hemorheology, coagulation function-related indicators, and histopathology to evaluate the replication of the blood stasis syndrome model and the effect of Yunnan Baiyao on promoting blood circulation and removing blood stasis, the biomarkers of blood stasis syndrome and related syndromes were identified through metabolomics technology. Metabolic pathways and the mechanism of Yunnan Baiyao for promoting blood circulation and removing blood stasis from the perspective of urine metabolomics were preliminarily explained. This study finally identified 28 potential biomarkers that had a significant impact on clustering. The increase or decrease in the content of these markers resulted in the metabolism disorder of the rat’s body. There were a total of eight metabolic pathways related to blood stasis syndrome by tracking the metabolic pathways related to changes in biomarkers.

Tyrosine and Phenylalanine Metabolic Pathway: Phenylpyruvate tautomerase, also called macrophage migration inhibitor or glycosylation inhibitor, participates in the metabolic pathways of tyrosine and phenylalanine and can catalyze the production of Enol-phenylpyruvate. 4-Hydroxyphenylacetaldehyde and the amino group of phosphatidylethanolamines in low-density lipoprotein in the intima of human atherosclerotic arteries form p-hydroxyphenylacetaldehyde-phosphatidylethanolamine, which is myeloperoxidase to lipid specific markers of qualitative damage [39]. Metabolic data analysis results showed that the content of 4-hydroxyphenylacetaldehyde in the urine metabolism of rats with blood stasis syndrome significantly increased, causing abnormal tyrosine metabolism and simultaneously leading to an increase in the content of p-hydroxyphenylacetaldehyde-phosphatidylethanolamine, resulting in lipid damage. 2,5-Dihydroxybenzaldehyde (Gentisate aldehyde) is a substrate of aldehyde oxidase I in tyrosine metabolism, and its significantly increased content causes tyrosine metabolism disorders. Normetanephrine is a metabolite of norepinephrine produced by the action of catechol-O-methyltransferase. It is a marker for catecholamine-secreting tumors such as pheochromocytoma [40], which has been studied. It is reported that blood stasis is one of the primary syndromes of pheochromocytoma [41]. Thus, the abnormal expression of methoxy norepinephrine may lead to the occurrence of blood stasis syndrome.

Taurine and hypotaurine metabolic pathways: Studies have found that high blood pressure is often accompanied by blood stasis syndrome [42,43], which is further developed based on syndromes such as hyperactive liver fire and deficiency of both Yin and Yang, causing the syndrome of clamping stasis. Lower doses of 5-L-Glutamyl-taurine can increase plasma renin concentration (PRC) and plasma renin activity (PRA) and lower blood pressure in a short time [44]. This study showed that the 5-L-glutamyltaurine content in the urine of model rats was significantly reduced. It is speculated that it may increase the plasma renin concentration (PRC) and plasma renin activity (PRA) expression, leading to increased blood pressure and blood stasis.

Arachidonic acid metabolic pathway: Scholars have confirmed that Yunnan Baiyao has a therapeutic effect on rheumatoid arthritis by regulating arachidonic acid metabolism in osteoblasts [45]. Studies have shown that the occurrence of blood stasis syndrome activates the protein kinase C pathway, leading to the phospholipids hydrolysis and the release of arachidonic acid. Arachidonic acid is also a substrate for the synthesis of certain bioactive compounds (eicosanoids), including prostaglandins, thromboxanes, and leukotrienes, which can act as mediators themselves and as modulators of other processes such as blood coagulates and smooth muscle contracts [46,47]. Prostaglandin E2 (PGE2) can increase vasodilation and cAMP production and enhance the effects of bradykinin and histamine [48,49]. Studies found that by regulating the levels of PGF2α, PGE2, and Ca2+ before menstruation, the uninhibited contraction of the uterine smooth muscle can be inhibited, and primary dysmenorrhea can be relieved [50]. In addition, PGE2 can significantly inhibit platelet aggregation and improve blood rheology indexes such as whole blood viscosity [49]. In this study, the PGE2 content of model rats decreased, which may increase platelet aggregation, leading to blood microcirculation disorders, and blood stasis [51].

Pyrimidine metabolic pathway: In this study, the content of Thymidine 5′-triphosphate and thymidine in model rats decreased, indicating that blood stasis syndrome caused abnormal pyrimidine metabolism. Studies reported that pyrimidine metabolism disorders could inhibit the synthesis of hypoxanthine transferase, resulting in hyperuricemia [52,53], accompanied by blood stasis [54].

Histidine metabolic pathway: Histidine can generate imidazole-4-acetaldehyde via histidine decarboxylase and histamine. Histidine has the effect of dilating blood vessels and is closely related to various allergies and inflammatory reactions [55]. In this study, the imidazole-4-acetaldehyde content increased, indicating that the blood stasis syndrome induced by low-dose epinephrine hydrochloride caused histidine metabolism disorder.

Lysine degradation and metabolic pathways: Pimelic acid derivatives are known to participate in lysine biosynthesis. Oxoadipic acid is a downstream metabolite of 2-aminoadipate. Studies have shown that 2-aminoadipic acid is one of the potential biomarkers for Qi deficiency and blood stasis syndrome, and Yang deficiency and water stop syndrome in patients with chronic heart failure [56]. In this study, the content of pimelic acid and 2-oxoadipate increased, which led to lysine metabolism disorder, resulting in blood stasis.

Other biomarkers and metabolic pathways: L-2-Hydroxyglutaric acid is converted into α-ketoglutaric acid by 2-hydroxyglutaric acid dehydrogenase. Studies have shown that the content changes in α-ketoglutarate, citric acid, taurine, and other components are significantly related to Qi deficiency and blood stasis syndrome [57].

## 5. Conclusions

In this study, an animal model of blood stasis syndrome related to the effects of Yunnan Baiyao for promoting blood circulation, relieving blood stasis, relieving pain, and reducing swelling was established, and the rationality of the model was verified using the measurement of the hemorheology, coagulation function, and histopathology of rats as indicators. In addition, based on the metabonomics method of UPLC-HDMS, the biomarkers and related metabolic pathways affected by Yunnan Baiyao were analyzed and explained. Yunnan Baiyao’s mechanism in activating blood circulation and removing blood stasis was revealed from urine metabolomics. Twenty-eight endogenous biomarkers were identified, targeting core metabolic pathways, including tyrosine metabolism, pyrimidine metabolism, phenylalanine metabolism, histidine metabolism, taurine, and hypotaurine metabolism arachidonic acid metabolism; tryptophan metabolism and lysine degradation are closely related to the formation of blood stasis syndrome. Through the biological network analysis tool, the network analysis of the key metabolites regulated by Yunnan Baiyao revealed that the arachidonic acid metabolic pathway in plasma may be an important pathway for Yunnan Baiyao to promote blood circulation and remove blood stasis. ALOX15 and EDN1 are the potential targets of Yunnan Baiyao for promoting blood circulation and removing blood stasis. In summary, ultra-high-performance liquid phase high-resolution mass spectrometry technology was used to discover abnormal metabolic pathways and analyze the target of Yunnan Baiyao for promoting blood circulation and removing blood stasis. Studies have shown that metabolomics has great potential in revealing the pathogenesis of TCM syndromes and effective targets of prescriptions. It also provides a new and effective way to accurately diagnose TCM syndromes and the mechanism of prescriptions [58,59].

## Figures and Tables

**Figure 1 molecules-27-03208-f001:**
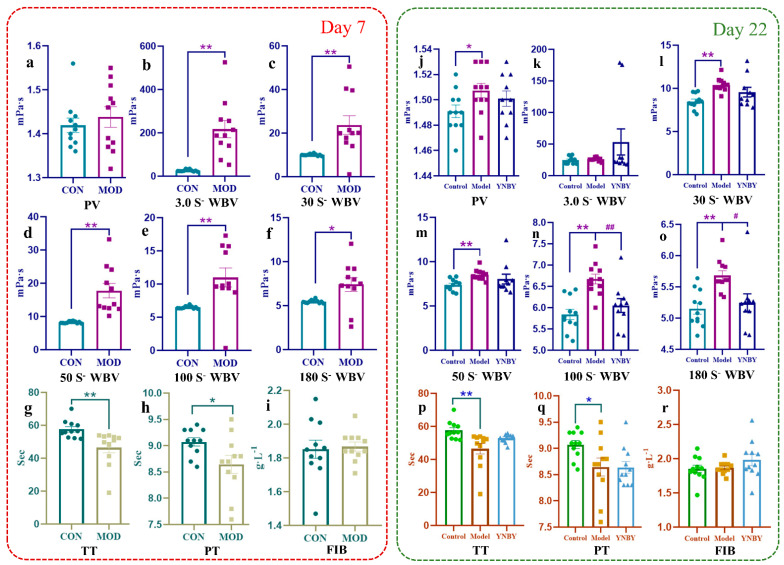
Evaluation of hemorheology and coagulation function. (**a**) Effect of Adr on PV of rats. (**b**–**f**) Shear rate is 3.0 S^−^, 30 S^−^, 50 S^−^,100 S^−^,180 S^−^, effect of Adr on CP of rats. (**g**–**i**) Effect of Adr on TT, PT, FIB of rats. (**j**) Effect of YNBY on PV of model rats. (**i**–**o**) Shear rate is 3.0 S^−^, 30 S^−^, 50 S^−^,100 S^−^, 180 S^−^, effect of YNBY on CP of model rats. (**p**–**r**) Effect of YNBY on TT, PT, FIB of model rats. * Significant differ-ence from control at *p* < 0.05. ** Significant difference from control at *p* < 0.01. # Significant dif-ference from YNBY at *p* < 0.05. ## Significant difference from YNBY at *p* < 0.01.

**Figure 2 molecules-27-03208-f002:**
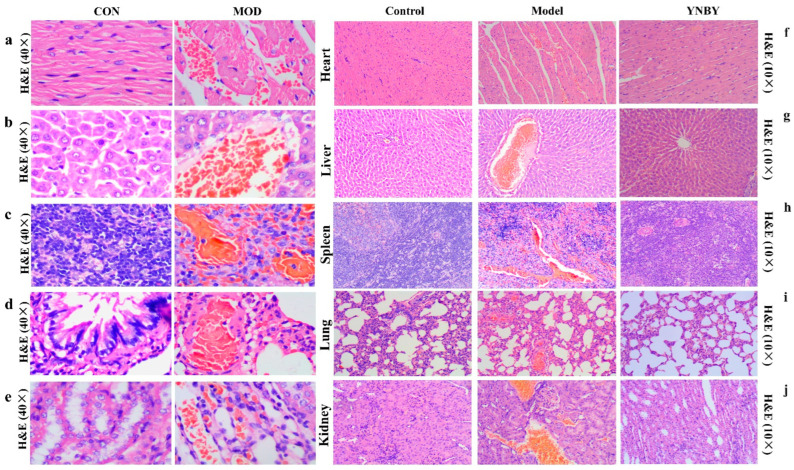
Histopathological evaluation. (**a**–**e**) HE staining of rat’s heart, liver, spleen, lung, kidney pathology (40×). (**f**–**j**) HE staining of rat’s heart, liver, spleen, lung, kidney pathology (10×).

**Figure 3 molecules-27-03208-f003:**
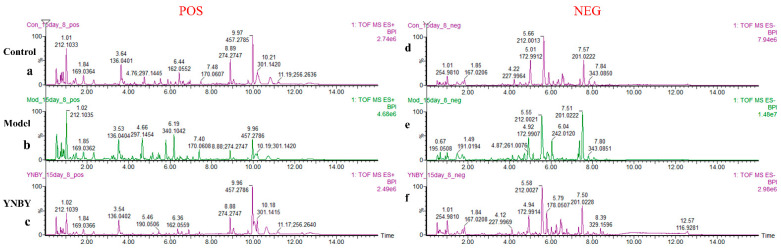
Metabolic fingerprints. UPLC–MS BPI serum chromatograms of each experimental group in positive mode and negative mode. (**a**,**d**): Control group; (**b**,**e**): Model group; (**c**,**f**): YNBY group. (**a**–**c**): Positive ionization mode; (**d**–**f**): Negative ionization mode.

**Figure 4 molecules-27-03208-f004:**
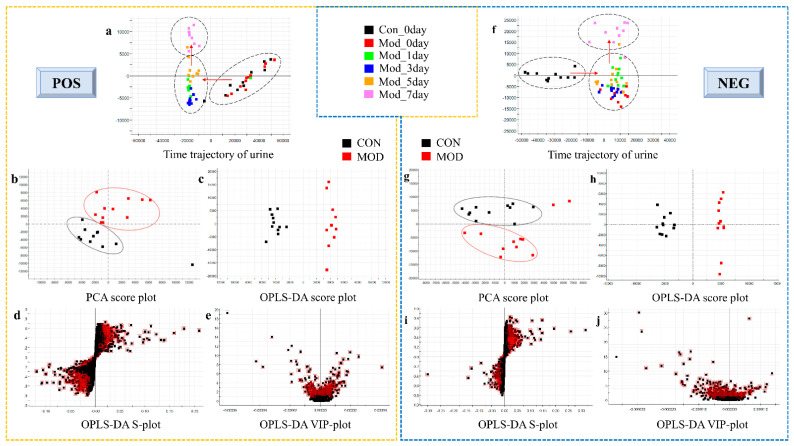
Multivariate statistical analysis of the UPLC–MS urine spectra data. (**a**,**f**) PCA score plot of urine metabolism of rats at different time points during modeling blood stasis syndrome. (**b**,**g**,**c**,**h**) PCA score plot and OPLS-DA score plot of urine metabolism on the 7th day. (**d**,**i**,**e**,**j**) S-plot and VIP-plot of urine metabolism on the 7th day. a and f: urine of control rats on 0 day; red dot: urine of model rats on 0 day; green dot: urine of model rats on 1 day; blue dot: urine of model rats on 3 day; orange dot: urine of model rats on 5 day; pink dot: urine of model rats on 7 day. (**b**,**c**,**g**,**h**): black dot: control group; red dot: model group.

**Figure 5 molecules-27-03208-f005:**
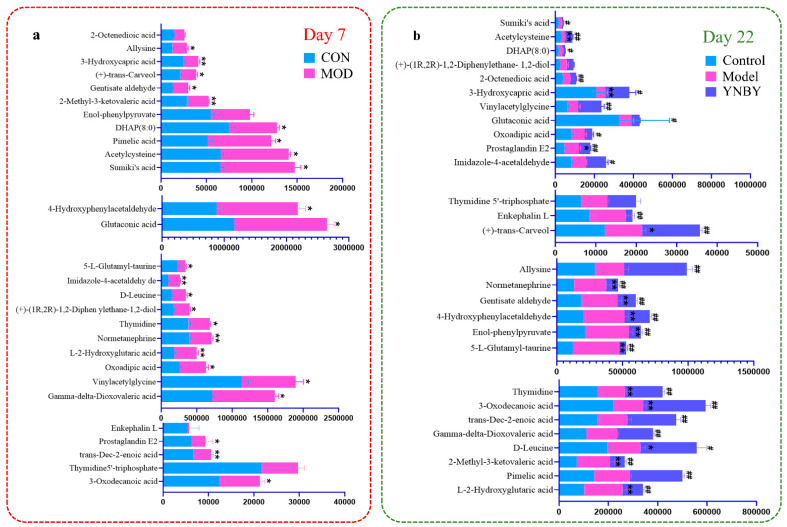
Relative signal intensities of metabolic biomarkers identified by UPLC–MS. (**a**): Bar plots represent the relative peak area ratios of 28 biomarkers. (**b**): YNBY has a 22/28 callback trend. Data are expressed as mean ± SD. * Significant difference from control at *p* < 0.05. ** Significant difference from control at *p* < 0.01. # Significant difference from YNBY at *p* < 0.05. ## Significant difference from YNBY at *p* < 0.01.

**Figure 6 molecules-27-03208-f006:**
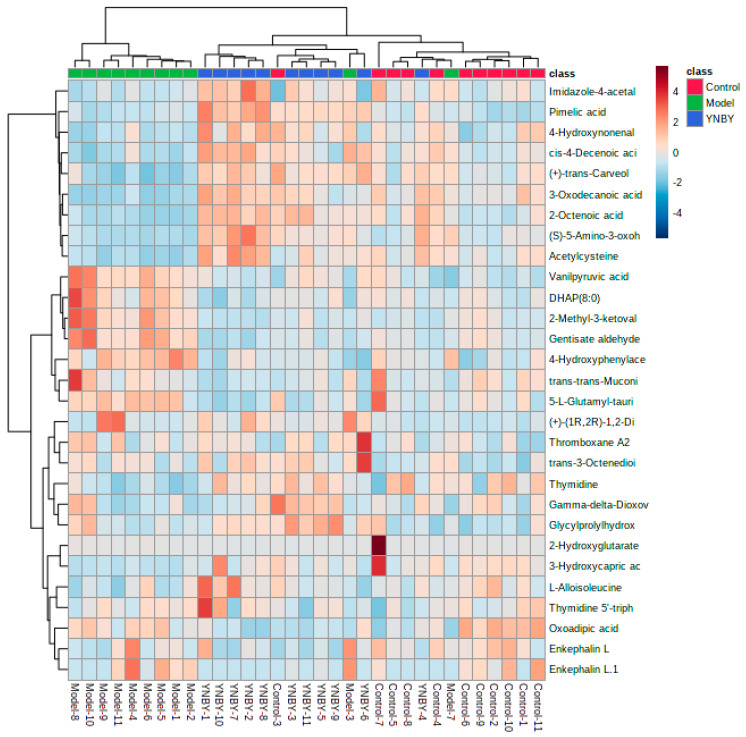
Heat map visualization for 28 biomarkers in urine from the control group, model group, and YNBY group.

**Figure 7 molecules-27-03208-f007:**
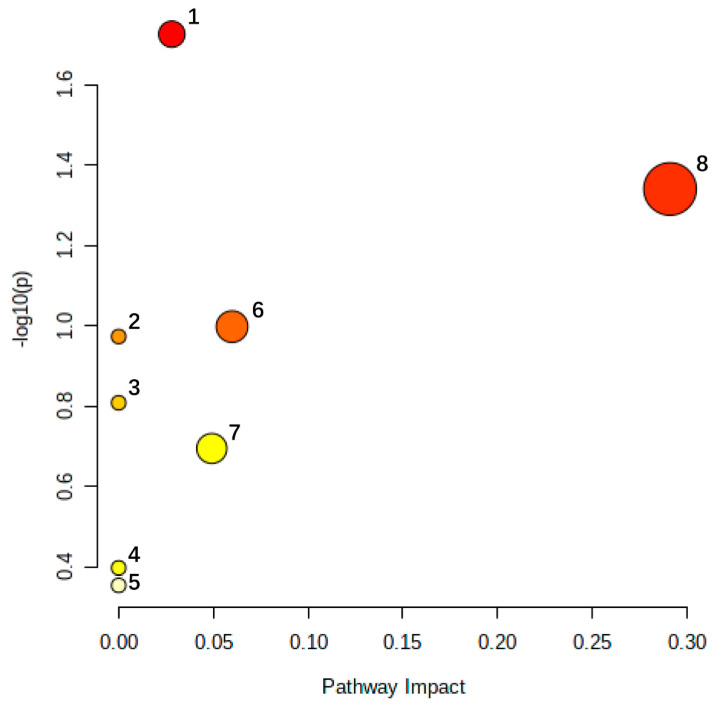
The metabolic pathway analysis of urine biomarkers in blood stasis syndromes rats. 1. Tyrosine metabolism; 2. Taurine and hypotaurine metabolism; 3. Phenylalanine metabolism; 4. Arachidonic acid metabolism; 5. Tryptophan metabolism; 6. Pyrimidine metabolism; 7. Histidine metabolism; 8. Lysine degradation.

**Figure 8 molecules-27-03208-f008:**
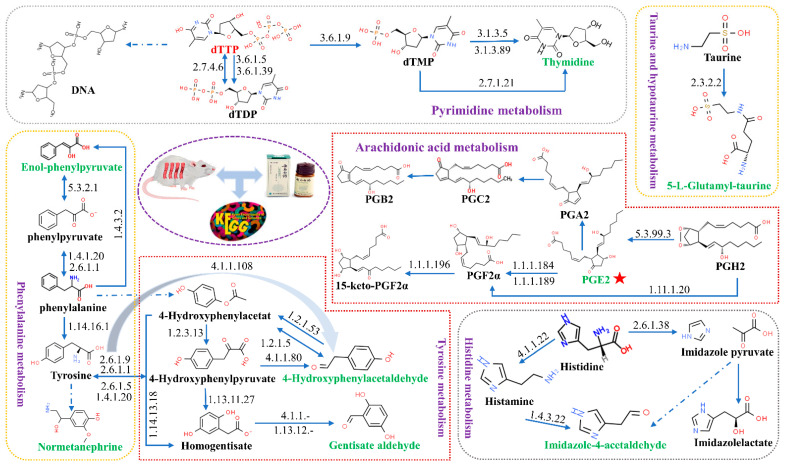
Urine metabolic pathways related to YNBY in treating blood stasis syndrome. Green text: the biomarkers influenced by YNBY; red text: the biomarkers that have not been called back by YNBY; black text: other components in the pathway; red star: key metabolite.

**Figure 9 molecules-27-03208-f009:**
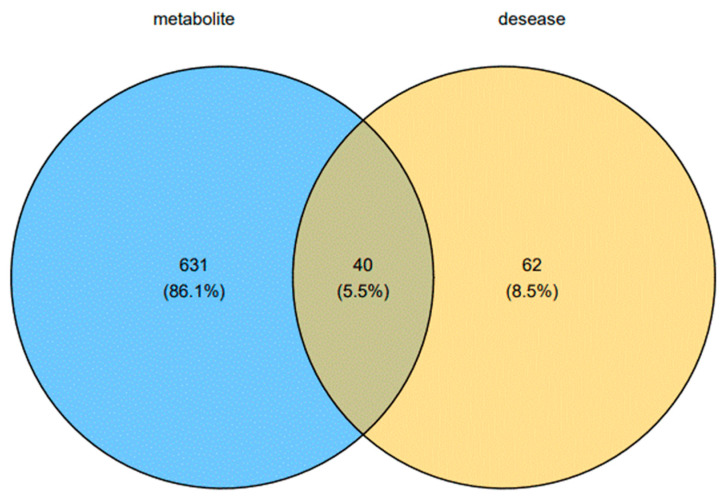
Venn diagrams of the metabolite targets and targets of hemorheological abnormality and coagulopathy.

**Figure 10 molecules-27-03208-f010:**
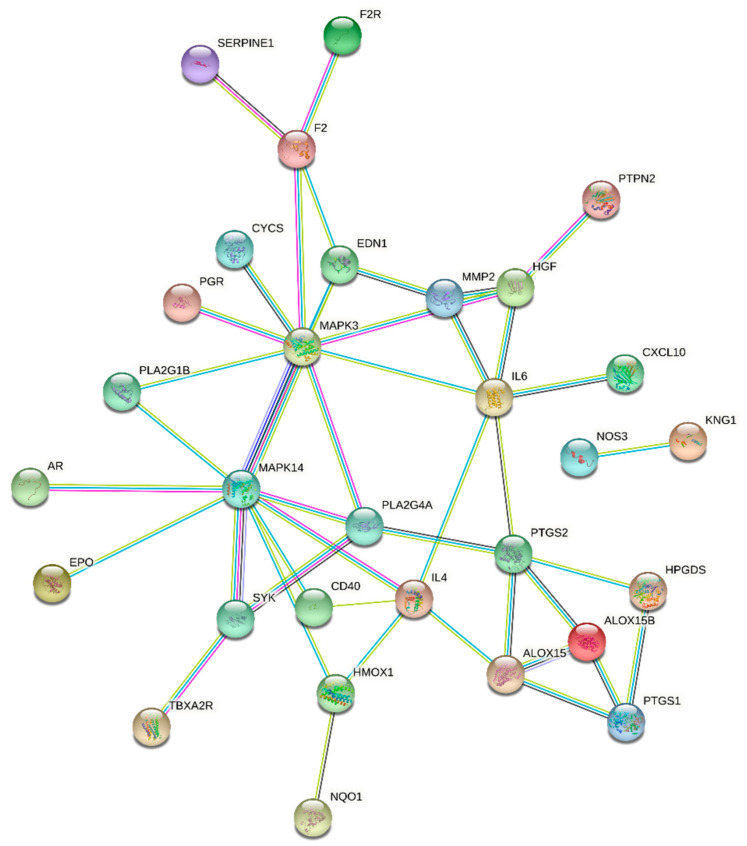
Protein–protein interaction network of the shared targets. F2: Coagulation Factor II; ALOX15: Arachidonate 15-Lipoxygenase; EDN1: Endothelin 1; MAPK14: Mitogen-Activated Protein Kinase 14; IL4: Interleukin 4; HGF: Hepatocyte Growth Factor; IL6: Interleukin 6; PTGS2: Prostaglandin-Endoperoxide Synthase 2; MAPK3: Mitogen-Activated Protein Kinase 3; PLA2G4A: Phospholipase A2 Group IVA.

**Table 1 molecules-27-03208-t001:** Information Table of Potential Biomarkers of Urine Metabolism.

NO.	Rt Min	*m*/*z* Determined	Adducts	HMDB	Proposed Composition	MS/MS Fragmentation (*m*/*z*)	Postulated Identity	Trend in Model
1	0.67	141.0190	M-H	HMDB0002432	C_6_H_6_O_4_	123.0335[M-H-H_4_O]^−^ 105.9658[M-H-H_5_O_2_]^−^ 80.9571[M-H-CH_2_O_3_]^−^	Sumiki’s acid	↑
2	0.97	155.0458	M + FA-H	HMDB0003905	C_5_H_6_N_2_O	93.0945[M-H-H_2_O]^−^ 79.9555[M-H-CH_3_O]^−^	Imidazole-4-acetaldehyde	↑
3	0.98	129.0188	M-H	HMDB0013233	C_5_H_6_O_4_	99.9185[M-H-CH_3_O]^−^ 85.0256[M-H-CH_2_O_2_]^−^	Gamma-delta-Dioxovaleric acid	↑
4	1.27	129.0189	M-H	HMDB0000620	C_5_H_6_O_4_	111.0621[M-H-H_4_O]^−^ 83.0652[M-H-CH_4_O_2_]^−^	Glutaconic acid	↑
5	2.12	159.0292	M-H	HMDB0000225	C_6_H_8_O_5_	115.0353[M-H-CH_2_O_2_]^−^ 98.0222[M-H-CH_3_O_3_]^−^	Oxoadipic acid	↑
6	2.37	147.0294	M-H	HMDB0000694	C_5_H_8_O_5_	103.0378[M-H-CH_2_O_2_] 87.0050[M-H-CH_2_O_3_]^−^ 85.0239[M-H-CH_4_O_3_]^−^	L-2-Hydroxyglutaric acid	↑
7	3.48	295.0947	M-H	HMDB0011685	C_11_H_21_O_7_P	253.0503[M-H-C_3_H_6_]^−^ 175.0756[M-H-C_3_H_5_O_3_P]^−^ 159.0454[M-H-C_4_H_9_O_3_P]^−^	DHAP (8:0)	↓
8	3.86	142.0503	M-H	HMDB0000894	C_6_H_9_NO_3_	124.0346[M-H-H_4_O]^−^ 98.0528[M-H-CH_2_O_2_]^−^ 83.0675[M-H-C_2_H_5_O_2_]^−^	Vinylacetylglycine	↓
9	4.02	144.0660	M-H	HMDB0001263	C_6_H_11_NO_3_	87.0020[M-H-C_2_H_5_NO]^−^ 85.0265[M-H- C_2_H_7_NO]^−^	Allysine	↑
10	4.26	242.0893	M-H	HMDB0000273	C_10_H_14_N_2_O_5_	203.0009[M-H-C_3_H_2_]^−^ 131.0824[M-H-C_5_H_4_NO_2_]^−^ 108.0213[M-H-C_5_H_11_NO_3_]^−^	Thymidine	↓
11	4.54	137.0238	M-H	HMDB0004062	C_7_H_6_O_3_	121.0161[M-H-H_2_O]^−^ 109.0313[M-H-CH_2_O]^−^	Gentisate aldehyde	↑
12	5.08	171.0655	M-H	HMDB0000341	C_8_H_12_O_4_	129.0916[M-H-CO_2_]^−^ 111.0816[M-H-C_2_H_4_O_2_]^−^ 86.9758[M-H-C_4_H_4_O_2_]^−^	2-Octenedioic acid	↓
13	5.29	554.2693	M-H	HMDB0001045	C_28_H_37_N_5_O_7_	236.1032[M-H-C_17_H_22_N_2_O_4_]^−^ 169.0723[M-H-C_22_H_29_N_2_O_4_]^−^ 130.0867[M-H-C_24_H_28_N_2_O_5_]^−^	Enkephalin L	↓
14	4.27	184.0963	M + H	HMDB0000819	C_9_H_13_NO_3_	166.1053[M+H-H_4_N]^+^ 150.1259[M+H-H_4_NO]^+^ 137.9975[M+H-CH_4_NO]^+^ 131.0403[M+H-CH_11_NO]^+^	Normetanephrine	↓
15	5.43	175.0606	M + FA-H	HMDB0000408	C_6_H_10_O_3_	96.9583[M-H-CH_6_O]^−^ 79.9693[M-H-C_2_H_11_O]^−^	2-Methyl-3-ketovaleric acid	↓
16	5.67	163.0397	M-H	HMDB0012225	C_9_H_8_O_3_	147.0202[M-H-H_2_O]^−^ 119.0432[M-H-CH_2_O_2_]^−^ 92.1066[M-H-C_2_HO_3_]^−^	Enol-phenylpyruvate	↓
17	5.70	135.0445	M-H	HMDB0003767	C_8_H_8_O_2_	136.0288[M-H-H]^−^ 108.0139[M-H-CHO]^−^ 93.0328[M-H-C_2_H_4_O]^−^	4-hydroxyphenylacetaldehyde	↑
18	5.79	159.0656	M-H	HMDB0000857	C_7_H_12_O_4_	144.0050 [M-H-HO]^−^ 125.0203[M-H-H_4_O_2_]^−^ 96.0089[M-H-CH_5_O_3_]^−^	Pimelic acid	↑
19	7.89	259.0985	M + FA-H	HMDB0012111	C_14_H_14_O_2_	137.0956[M-H-C_6_H_6_]^−^ 135.0413[M-H-C_6_H_8_]^−^	(+)-(1R,2R)-1,2-Diphen ylethane-1,2-diol	↑
20	6.39	253.0504	M-H	HMDB0004195	C_7_H_14_N_2_O_6_S	209.0609[M-H-CO_2_]^−^ 165.1283[M-H-C_2_H_2_NO_3_]^−^ 143.0500[M-H-C_2_H_8_NO_4_]^−^	5-L-Glutamyl-taurine	↓
21	6.40	480.9840	M-H	HMDB0001342	C_10_H_17_N_2_O_14_P_3_	242.0126[M-H-H_2_O_9_P_3_]^−^ 212.0024[M-H-CH_4_O_10_P_3_]^−^ 162.0537[M-H-C_3_H_14_O_11_P_3_]^−^	Thymidine 5′-triphosphate	↓
22	6.47	130.0867	M-H	HMDB0013773	C_6_H_13_NO_2_	89.0084[M-H-C_3_H_7_]^−^ 72.9893[M-H-C_3_H_7_O]^−^	D-Leucine	↑
23	7.51	185.1135	M-H	HMDB0010724	C_10_H_18_O_3_	141.0911[M-H-C_3_H_8_]^−^ 125.1014[M-H-C_3_H_8_O]^−^ 109.0276[M-H-C_4_H_12_O]^−^ 79.9580[M-H-C_5_H_13_O_2_]^−^	3-Oxodecanoic acid	↓
24	8.06	187.1332	M-H	HMDB0002203	C_10_H_20_O_3_	151.0831[M-H-H_6_O_2_]^−^ 145.0560[M-H-C_2_H_4_O]^−^ 124.0039[M-H-C_2_H_9_O_2_]^−^	3-Hydroxycapric acid	↓
25	8.25	351.2160	M-H	HMDB0001220	C_20_H_32_O_5_	351.2107[M-H]^−^ 162.0184[M-H-C_10_H_21_O_3_]^−^ 115.0025[M-H-C_14_H_20_O_3_]^−^	Prostaglandin E2	↓
26	8.25	151.1121	M-H	HMDB0059608	C_10_H_16_O	136.0390[M-H-HO]^−^ 122.0321[M-H-CH_3_O]^−^ 94.0446[M-H-C_3_H_7_O]^−^	(+)-trans-Carveol	↓
27	8.57	215.1281	M+FA-H	HMDB0010726	C_10_H_18_O_2_	140.0370[M-H-C_2_H_7_]^−^ 127.0250[M-H-C_3_H_8_]^−^ 97.0009[M-H-C_5_H_14_]^−^ 83.0496[M-H-C_6_H_16_]^−^	trans-Dec-2-enoic acid	↓
28	8.75	162.0226	M-H	HMDB0001890	C_5_H_9_NO_3_S	147.0233[M-H-HO]^−^ 133.0462[M-H-CH_3_O]^−^ 102.0408[M-H-C_2_H_6_O_2_]^−^	Acetylcysteine	↑

The arrows in the table show relative change trends of potential biomarkers compared with the model group. ↑ indicates overexpressed; ↓ indicates insufficiently expressed.

## Data Availability

The data presented in this study are available in the Appendix A here.

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
