# Peer review of "UPLC-G2Si-HDMS Untargeted Metabolomics for Identification of Yunnan Baiyao’s Metabolic Target in Promoting Blood Circulation and Removing Blood Stasis"

_molecules, 2022, doi:10.3390/molecules27103208_

Round 1

Reviewer 1 Report

In this manuscript, authors demonstrates that Yunnan Baiyao was found to be effective in promoting blood circulation, reducing blood stasis, relieving pain, and reducing swelling in an animal model of blood stasis, and the model's rationality was tested by measuring hemorheology, coagulation function, and histopathology of rats as indicators. Additionally, Yunnan Baiyao was analyzed and explained as a result of the metabonomics method of UPLC-HDMS

        This manuscript is well written and easy to read. Additionally, the results are interesting, but some issues should be solved. I think this manuscript can be published in "molecules" after revision. These are the followings:

Comments:

  1. Authors must provide the proof from Institutional Animal Care and Use Committee, IACUC.
  2. In the figure 2, the scar bar was missing.
  3. In the figure 6, heat map should be corrected to Heat map. Additionally, the candidate genes must be selected more than two candidates and further verified by real-time PCR or western blot.
  4. Please remove the Appendix A in this manuscript.
  5. Before for publication, authors should consult an English language editor and write in an "academic English" as well as "must" provide proof of a certificate of editing.

Author Response

Journal Molecules (ISSN 1420-3049)

Manuscript ID molecules-1697256

Type Article

Title UPLC-G2Si-HDMS untargeted metabolomics for identification of Yunnan Baiyao's metabolic target in promoting blood circu-lation and removing blood stasis

Authors

Qing-yu Zhang , Ai-hua Zhang , Fang-fang Wu , Xi-jun Wang *

Section

Medicinal Chemistry

Special Issue

Natural Compounds: A Lead for Drug Discovery and Development

In this manuscript, authors demonstrates that Yunnan Baiyao was found to be effective in promoting blood circulation, reducing blood stasis, relieving pain, and reducing swelling in an animal model of blood stasis, and the model's rationality was tested by measuring hemorheology, coagulation function, and histopathology of rats as indicators. Additionally, Yunnan Baiyao was analyzed and explained as a result of the metabonomics method of UPLC-HDMS

This manuscript is well written and easy to read. Additionally, the results are interesting, but some issues should be solved. I think this manuscript can be published in "molecules" after revision. These are the followings:

Comments:

  1. Authors must provide the proof from Institutional Animal Care and Use Committee, IACUC.

Answer: Due to the impact of the "New Coronary Pneumonia" epidemic, the GLP institution of Heilongjiang University of Traditional Chinese Medicine is in a state of shutdown management, no management personnel are on duty, and the IACUC certificate cannot be retrieved. However, the animal study was reviewed and approved by the ethics of Heilongjiang University of Chinese Medicine. Permit number: SYXK (HL) 2020-004.

  1. In the figure 2, the scar bar was missing.

Answer: In the model evaluation, pathological pictures under a 40x microscope were selected to facilitate readers to compare the degree of organ damage caused by model replication; pathological pictures under a 10x microscope were used in drug treatment to facilitate readers to understand the degree of organ damage in model rats based on, expand the field of view of pathological pictures, highlight the effect of contrast drug treatment, trouble the editor to review again.

  1. In the figure 6, heat map should be corrected to Heat map. Additionally, the candidate genes must be selected more than two candidates and further verified by real-time PCR or western blot.

Answer: The Heat map has been changed, thanks to the editor's careful reminder, for "the candidate genes must be selected more than two candidates and further verified by real-time PCR or western blot.", under the state of epidemic shutdown and limited time, we have adopted a simple Fast alternative method, Through the biological network analysis tool, the network analysis of the key metabo-lites regulated by Yunnan Baiyao revealed that the arachidonic acid metabolic pathway in plasma may be an important pathway for Yunnan Baiyao to promote blood circulation and remove blood stasis.  ALOX15 and EDN1 are the potential targets of Yunnan Baiyao for promoting blood circulation and removing blood stasis. With the help of published disease target validation studies, we focus on the targets of Yunnan Baiyao in this study to promote blood circulation and remove blood stasis, please editor's approval.

  1. Please remove the Appendix A in this manuscript.

Answer: Appendix A in this manuscript has been removed as requested.

  1. Before for publication, authors should consult an English language editor and write in an "academic English" as well as "must" provide proof of a certificate of editing.

Reviewer 2 Report

Dear Editor and Authors,

The manuscript ‘UPLC-G2Si-HDMS untargeted metabolomics for identification  of Yunnan Baiyao's metabolic target in promoting blood circulation and removing blood stasis’ by Qing-yu Zhang, Ai-hua Zhang, Fang-fang Wu, Xi-jun Wang describes research on rats. Rats had epinephrine hydrochloride induced blood stasis syndrome’ and results were compared with control rats. The Authors proven that Yunnan Baiyao was able to prevent blood stasis syndrome. They used a non targeted metabolomics method based on high-resolution liquid chromatography-quadru- pole time-of-flight mass spectrometry (UPLC-QTOF-MS) to study the endogenous metabolism of Yunnan Baiyao in rat urine.

The manuscript needs minor revision.

Abstract

Line 20 Please explain in what sense the rats were different from ‘normal rats’

Maybe something like that:

‘Rats with epinephrine hydrochloride induced blood stasis syndrome’

Line 38

I do not like the sentence ‘It is widely used in treating various bleeding and blood stasis diseases in different departments such  as internal medicine, external medicine, women, children, five sense organs, skin [9, 10], all of which achieved satisfactory therapeutic effects.’

I suggest something like that:

‘It is widely  used in treating various bleeding and blood stasis diseases both in internal medicine and external medicine, especially in curing/preventing women’s, and children’s diseases, in curing/preventing five sense organs, and skin diseases [9, 10], all of which achieved satisfactory therapeutic effects.’

Line 48 Again, I do not appreciate: ‘ in various clinical diseases, including internal, external, women, children, skin in various departments such as the five sense organs, nerves, and tumors’. Please, do not call women a disease!

Maybe something like this: ‘various clinical diseases, including internal, external bleeding; in women, and children; in diseases of the skin, the five sense organs, nerves, and even tumors.

Line 82

Sprague–Dawley (SD)

Line 91

The CON and MOD groups were sacrificed on the 8th day for  model evaluation. ‘Sacrificed’ means killed, so how could they were given anything? ‘The treatment group was given an equivalent dose of Yunnan Baiyao  solution (180 mg/kg) by gavage on the 8th day. The other groups of rats were given distilled water daily by gavage for 15 consecutive days.’?

Please explain.

How many rats were in each group.

Line 97 give ethical commission permission number

Line 173 ‘After seven consecutive days, compared with the control group, the rats in the model group showed signs of reduced mobility, they curled up and moved less, were drowsy, and had a dull coat.’

Your sincerely,

Author Response

The manuscript ‘UPLC-G2Si-HDMS untargeted metabolomics for identification  of Yunnan Baiyao's metabolic target in promoting blood circulation and removing blood stasis’ by Qing-yu Zhang, Ai-hua Zhang, Fang-fang Wu, Xi-jun Wang describes research on rats. Rats had epinephrine hydrochloride induced blood stasis syndrome’ and results were compared with control rats. The Authors proven that Yunnan Baiyao was able to prevent blood stasis syndrome. They used a non targeted metabolomics method based on high-resolution liquid chromatography-quadru- pole time-of-flight mass spectrometry (UPLC-QTOF-MS) to study the endogenous metabolism of Yunnan Baiyao in rat urine.

The manuscript needs minor revision.

Abstract

Line 20 Please explain in what sense the rats were different from ‘normal rats’

Maybe something like that:

‘Rats with epinephrine hydrochloride induced blood stasis syndrome’

Answer: The modification has been completed according to the editor's suggestion, please check again.

Line 38

I do not like the sentence ‘It is widely used in treating various bleeding and blood stasis diseases in different departments such  as internal medicine, external medicine, women, children, five sense organs, skin [9, 10], all of which achieved satisfactory therapeutic effects.’

I suggest something like that:

‘It is widely  used in treating various bleeding and blood stasis diseases both in internal medicine and external medicine, especially in curing/preventing women’s, and children’s diseases, in curing/preventing five sense organs, and skin diseases [9, 10], all of which achieved satisfactory therapeutic effects.’

Answer: The modification has been completed according to the editor's suggestion, please check again.

Line 48 Again, I do not appreciate: ‘ in various clinical diseases, including internal, external, women, children, skin in various departments such as the five sense organs, nerves, and tumors’. Please, do not call women a disease!

Maybe something like this: ‘various clinical diseases, including internal, external bleeding; in women, and children; in diseases of the skin, the five sense organs, nerves, and even tumors.

Answer: The modification has been completed according to the editor's suggestion, please check again.

Line 82

Sprague–Dawley (SD)

Answer: The modification has been completed according to the editor's suggestion, please check again.

Line 91

The CON and MOD groups were sacrificed on the 8th day for  model evaluation. ‘Sacrificed’ means killed, so how could they were given anything? ‘The treatment group was given an equivalent dose of Yunnan Baiyao  solution (180 mg/kg) by gavage on the 8th day. The other groups of rats were given distilled water daily by gavage for 15 consecutive days.’?

Answer: In the experimental design, because of the inclusion of other experimental contents, in order to meet the needs of biological samples such as urine and blood during the experiment, acontrol model accompanying group was set up to prepare for sufficient samples.

Please explain.

How many rats were in each group.

Answer: There were 11 mice in each group.

Line 97 give ethical commission permission number

Answer: The animal study was reviewed and approved by the ethics committee of Heilongjiang University of Chinese Medicine. Permit number:SYXK(HL)2020-004.

Line 173 ‘After seven consecutive days, compared with the control group, the rats in the model group showed signs of reduced mobility, they curled up and moved less, were drowsy, and had a dull coat.’

Answer: Yes, after seven consecutive days, through behavioral observation, it was found that the rats in the blood stasis model group had clinical manifestations of blood stasis syndrome.

Round 2

Reviewer 1 Report

Dear authors,

Many thanks for your revision.